

# Identifying communities from multiplex biological networks

Gilles Didier[1], Christine Brun[2,3] and Anaïs Baudot[1]

[1] Aix Marseille Université, CNRS, Centrale Marseille, I2M UMR 7373, Marseille, France
[2] Aix Marseille Université, Inserm, TAGC UMR_S1090, Marseille, France
[3] CNRS, Marseille, France

## ABSTRACT

Various biological networks can be constructed, each featuring gene/protein relationships of different meanings (e.g., protein interactions or gene co-expression). However, this diversity is classically not considered and the different interaction categories are usually aggregated in a single network. The multiplex framework, where biological relationships are represented by different network layers reflecting the various nature of interactions, is expected to retain more information. Here we assessed aggregation, consensus and multiplex-modularity approaches to detect communities from multiple network sources. By simulating random networks, we demonstrated that the multiplex-modularity method outperforms the aggregation and consensus approaches when network layers are incomplete or heterogeneous in density. Application to a multiplex biological network containing 4 layers of physical or functional interactions allowed recovering communities more accurately annotated than their aggregated counterparts. Overall, taking into account the multiplexity of biological networks leads to better-defined functional modules. A user-friendly graphical software to detect communities from multiplex networks, and corresponding C source codes, are available at GitHub (https://github.com/gilles-didier/MolTi).

Corresponding author
Gilles Didier, gilles.didier@univ-amu.fr
Anaïs Baudot, anais.baudot@univ-amu.fr

## INTRODUCTION

Biological macromolecules do not act in isolation, but rather interact with each other to perform their cellular functions. In particular, thousands of interactions are observed between proteins, forming the basis of cellular biological processes. Thanks to the scaling of the experimental techniques allowing interaction discovery, and to the development of centralized databases, recent years have witnessed the accumulation of thousands of physical and functional interactions of various nature (*Sharan & Ideker, 2006*). For instance, data obtained from yeast two-hybrid and affinity purification-mass spectrometry experiments inform on the physical interactions between proteins and their organization as molecular complexes; mRNA expression correlations identify the genes that are co-expressed in a set of experimental conditions, and signaling pathway data describe the cascades of reactions transmitting signals within and between cells.

Interaction data are usually represented as networks, i.e., graphs in which vertices correspond to genes or proteins, and edges to interactions. One of the leading approaches to extract functional knowledge from biological networks is their clustering into communities—or modules—of tightly linked genes/proteins (*Aittokallio & Schwikowski, 2006*). These modules are expected to represent the building blocks of the cells (*Hartwell et al., 1999*). Clustering algorithms developed for community detection are frequently based on graph topological properties, such as density or modularity (for reviews, *Wang et al., 2010*; *Pizzuti & Rombo, 2014*). They have been thoroughly assessed on biological networks (*Brohée & Van Helden, 2006*), and are widely used in biology (*Arroyo et al., 2015*; *Wan et al., 2015*; *Huttlin et al., 2015*; *Chapple et al., 2015*; *Katsogiannou et al., 2014*). However, they have been applied on single networks whereas recently *multiplex* (alt. *multi-layer* or *multi-slice*) networks (*Kivelä et al., 2014*) have been introduced in biology. These multiplex networks are sets of networks sharing the same nodes, but in which edges belong to different categories or represent interactions of different nature. As biological interaction sources are diverse, biological multiplex networks contain sparse layers of high relevance (e.g., curated signaling pathway networks), while other layers contain thousands or even millions of interactions (e.g., co-expression networks). In order to extract relevant biological information, it is thus important to assess the approaches able to detect communities from these multiplex biological networks.

As the majority of current clustering approaches takes as input a single network, the interactions from the different sources are classically aggregated into a unique network, herein called *monoplex*, in which all the edges are equally considered, regardless of their molecular nature. For instance, in the chaperome network (*Brehme et al., 2014*), protein–protein interactions and co-expression associations were aggregated and the link-community clustering algorithm (*Ahn, Bagrow & Lehmann, 2010*) was applied on the resulting monoplex network to identify communities. However, considering all the interaction categories as equivalent likely excludes important information, because the gene/protein relationships have different meanings or relevance (*Battiston, Nicosia & Latora, 2014*). On another hand, approaches known as *consensus clustering* aggregate *a posteriori* the communities obtained from several independent networks (*Senbabaoglu, Michailidis & Li, 2014*; *Lancichinetti & Fortunato, 2012*). Finally, more recently, different topological measures have been adapted to multiplex networks (*Kivelä et al., 2014*; *Battiston, Nicosia & Latora, 2014*), and a handful of community detection algorithms can take multiple networks as input (*Mucha et al., 2010*; *Shiga & Mamitsuka, 2012*; *Papalexakis, Akoglu & Ience, 2013*; *Bennett et al., 2015*). To our knowledge, these 3 sets of approaches, namely network aggregations, consensus clustering and multiplex approaches, have not been extensively compared and evaluated.

In order to do so, we carefully explored modularity-based approaches by identifying communities from multiplex networks using (i) the union-, intersection-, or sum-aggregations of the layers of the multiplex network, (ii) the consensus of the community obtained on the individual layers and (iii) a natural extension of the modularity, called here multiplex-modularity, that allows identifying communities directly from multiplex networks (similar to *Bennett et al. (2015)*). Importantly, the strong assumption that all
the network sources share the same community structure underlies all these approaches. More generally, this assumption is implicit for any approach returning a single community structure from a multiplex network.

We investigated whether combining multiple network sources could improve the detection of communities, in particular when noise impairs the inference of the community structure from individual networks. More particularly, we assessed the influence of important features characterizing biological networks, such as their heterogeneous densities and the weight of missing data. We first ran extensive simulations on random multiplex networks generated with a community structure, and mimicking the sparsity, the heterogeneity and the incompleteness of biological networks. We then showed that (i) the aggregation of the layers of the multiplex networks through their union and intersection ultimately erases the community structure, and (ii) the multiplex-modularity clustering outperforms the aggregation and consensus approaches. Importantly, the multiplex-modularity approach is more reliable than all other approaches when the set of studied networks are incomplete or heterogeneous in density, i.e., when networks display features of real biological networks.

We then studied a real-case of multiplex biological network composed of 4 interaction layers. Communities obtained with the multiplex-modularity are more associated to significantly enriched biological processes compared to their aggregated counterparts. We detailed, as an example, a module that clusters the proteins implicated in the Coffin-Siris syndrome. These proteins are clustered with proteins involved in other syndromes displaying overlapping clinical features.

Finally, we provide *MolTi*, a standalone graphical interface allowing the users to cluster multiple input networks, annotate the obtained clusters and parse the resulting communities.

## MATERIAL AND METHODS

### Multiplex networks, modularity and community detection
#### Monoplex and multiplex networks

Let $X$ be a (monoplex) network, i.e., an undirected graph. We use $X$ for its incidence matrix too, where the entry $X_{i,j}$ indicates the presence ($X_{i,j} = 1$) or the absence ($X_{i,j} = 0$) of an edge between the vertices $i$ and $j$.

A *multiplex network* is a collection of undirected graphs $(X^{(g)})_g$ sharing the same set of vertices (*Kivelä et al., 2014*). A basic way to apply standard graph approaches on a given multiplex network is to first aggregate all its graph layers into a monoplex network. We considered here three types of aggregation, the intersection $X^{\cap}$, the union $X^{\cup}$ and the sum $X^{\Sigma}$ of the graphs in $(X^{(g)})_g$, respectively defined as:

$$X_{i,j}^{\cap} = \bigwedge_g X_{i,j}^{(g)}, X_{i,j}^{\cup} = \bigvee_g X_{i,j}^{(g)} \text{ and } X_{i,j}^{\Sigma} = \sum_g X_{i,j}^{(g)}$$

for all pairs of vertices $i$ and $j$, where $\wedge$ and $\vee$ denote the logical conjunction and the (inclusive) disjunction respectively. Note that $X^{\cap}$ and $X^{\cup}$ are unweighted graphs whereas $X^{\Sigma}$ is a weighted graph.

### Modularity

Under the assumption that a network $X$ has a community structure $\mathbf{c}$ and by putting $\mathbf{c}_i$ for the community of the vertex $i$, the Newman–Girvan modularity (*Newman & Girvan, 2004*) is defined as:

$$Q(X, \mathbf{c}) = \frac{1}{2m} \sum_{\substack{\{i,j\} \\ i \neq j}} \left( X_{i,j} - \frac{k_i k_j}{2m} \right) \delta_{\mathbf{c}_i, \mathbf{c}_j} \tag{1}$$

$$= \sum_a \left[ \frac{\sum\limits_{\substack{\{i,j\} \\ i \neq j, \mathbf{c}_i = \mathbf{c}_j = a}} X_{i,j}}{2m} - \frac{\sum\limits_{\substack{\{i,j\} \\ i \neq j, \mathbf{c}_i = \mathbf{c}_j = a}} k_i k_j}{(2m)^2} \right] \tag{2}$$

where $m$ is the total number of edges of $X$, $k_i$ is the number of edges containing the vertex $i$ (i.e., the degree of $i$) and $\delta_{\mathbf{c}_i, \mathbf{c}_j} = \begin{cases} 1 & \text{if } \mathbf{c}_i = \mathbf{c}_j \\ 0 & \text{otherwise} \end{cases}$.

The modularity was initially conceived as a measure of the strength of a community structure (*Newman & Girvan, 2004*). In the original article, it is defined as the sum over all the communities $a$, of the proportions of within-community edges (first term of Eq. (2)) minus the expectation of these proportions in random networks with the same vertex degrees (second term of Eq. (2)).

This modularity has been extended to deal with weighted graphs, by considering the sum of the weights instead of the number of edges (*Newman, 2004*).

### Multiplex-modularity

In order to extend the modularity of *Newman & Girvan (2004)* to multiplex networks, we measured the strength of a given community structure in all the graphs of a multiplex network $(X^{(g)})_g$ by considering, for all communities $a$, the sum of the proportions of within-community edges over all the graphs minus the expectation of this sum under the same model as in *Newman & Girvan (2004)*. Since the expectation of the sum of random variables is the sum of their expectations, we obtained the Eq. (3). Reorganizing this sum shows that it is equal to the sum of the individual modularities of the graphs of the multiplex with regard to the same community partition Eq. (5). The *multiplex-modularity* of the multiplex network $(X^{(g)})_g$ is defined as:

$$Q^M((X^{(g)})_g, \mathbf{c}) = \sum_a \left[ \sum_g \frac{\sum\limits_{\substack{\{i,j\} \\ i \neq j, \mathbf{c}_i = \mathbf{c}_j = a}} X_{i,j}^{(g)}}{2m^g} - \sum_g \frac{\sum\limits_{\substack{\{i,j\} \\ i \neq j, \mathbf{c}_i = \mathbf{c}_j = a}} k_i^g k_j^g}{(2m^g)^2} \right] \tag{3}$$

$$= \sum_g \frac{1}{2m^g} \sum_{\substack{\{i,j\} \\ i \neq j}} \left( X_{i,j}^{(g)} - \frac{k_i^g k_j^g}{2m^g} \right) \delta_{\mathbf{c}_i, \mathbf{c}_j} \tag{4}$$

$$= \sum_g Q(X^{(g)}, \mathbf{c}) \tag{5}$$

where $m^g$ is the total number of edges of the graph $X^{(g)}$ and $k_i^g$ is the degree of the vertex $i$ in the graph $X^{(g)}$.

The multiplex-modularity could easily be adapted to multiplex networks containing weighted edges.

### Clustering algorithms for community detection

The network modularity was initially designed to measure the quality of a partition into communities (*Newman & Girvan, 2004*). Several clustering algorithms were subsequently developed to identify the communities by optimizing the modularity (*Blondel et al., 2008*; *Shiokawa, Fujiwara & Onizuka, 2013*). Finding the partition optimizing the modularity is NP-complete (*Brandes et al., 2008*) and thus requires meta-heuristic approaches when dealing with large graphs such as biological networks. We applied the Louvain algorithm (*Blondel et al., 2008*). The Louvain algorithm starts from the community structure that separates all vertices. Next, it tries to move each vertex from its community to another, keeping only the changes increasing the modularity, until no change increases the modularity. It then replaces the vertices by the detected communities and iterates the same operations on the new graph obtained, until stability. This algorithm was applied to the networks aggregated as $X^{\cap}$, $X^{\cup}$ and $X^{\Sigma}$. Adapting Louvain to the multiplex networks required replacing the optimization of the modularity by the optimization of the multiplex-modularity defined previously.

### Resolution parameter

A general issue with modularity-based clustering approaches is that they may fail to detect communities smaller than a certain size, depending on the graph size. This phenomena is known as the *resolution limit* (*Fortunato & Barthélemy, 2007*). This issue is overcame by adding a *resolution parameter* $\gamma$ to the modularity formula (*Reichardt & Bornholdt, 2006*). The *parametrized modularity* $\mathcal{Q}_\gamma$ is defined as:

$$\mathcal{Q}_\gamma(X, \mathbf{c}) = \frac{1}{2m} \sum_{\substack{\{i,j\} \\ i \neq j}} \left( X_{i,j} - \gamma \frac{k_i k_j}{2m} \right) \delta_{\mathbf{c}_i, \mathbf{c}_j}. \tag{6}$$

The greater the resolution parameter $\gamma$, the smaller the communities maximizing the corresponding modularities. The *parametrized multiplex-modularity* $\mathcal{Q}_\gamma^M$ of a multiplex network is defined accordingly.

### Consensus clustering

Consensus clustering approaches compute a unique community structure from the community structures obtained on each graph independently. We implemented a consensus clustering approach inspired from *Lancichinetti & Fortunato (2012)*. It first computes the community structures of each individual graph with the standard Louvain algorithm. Next, each community structure is transformed into a new graph in which an edge connects two vertices if they belong to the same community. We obtained the same number of new graphs as the initial number of graphs composing the multiplex network. These new graphs form a new multiplex network from which we infer a single community structure by using the sum-aggregation approach and the Louvain algorithm.

## Simulated multiplex networks
### Random graph models
Stochastic block models (SBMs, also known as planted partition models) are among the most intuitive models of random graphs with community structures (*Holland, Laskey & Leinhardt, 1983*). In SBMs, the number of communities is set *a priori* and each vertex $i$ is assigned to a unique community $c_i$. The presence of an edge between two vertices $i$ and $j$ is drawn independently conditionally on the respective community assignments of $i$ and $j$, and with a probability that only depends on these assignments. Note that edges are independent conditionally on their communities, although not independent in an absolute sense. For instance, two edges involving vertices of the same community are more likely to be both present than edges from different communities. Hence, the parameters of the SBMs are the number of vertices, the community structure and the matrix $\pi$ in which the entry $\pi_{ql}$ is the probability of observing an edge between a vertex of the community $q$ and one of the community $l$.

A natural way to generate multiplex networks $(X^{(g)})_g$ with a community structure is to draw independent realizations of SBMs sharing the same community structure. This means that all the graphs are drawn independently and, in each graph, all the edges are also drawn independently, conditionally on the community assignments of the vertices. The parameters of the multiplex SBMs are the same as classical SBMs, except that instead of one matrix of edge probabilities, we have a collection $(\pi^{(g)})_g$, the $g$th matrix corresponding to the $g$th graph of the multiplex network. To ease the interpretation of the simulations, we parametrized edge probability matrices with only two values: the probability $p_I$ of an internal edge and the probability $p_E$ of an external one. All diagonal entries are equal to $p_I$ and all non-diagonal entries are equal to $p_E$.

### Network incompleteness and missing data
We simulated missing data by setting an arbitrary probability for a vertex to be absent in a graph, and using it to randomly remove some vertices. The community detection algorithm was then applied to the incomplete multiplex networks. In practice, whenever a vertex is missing in a graph layer, it is considered as not interacting in this graph.

## Biological multiplex networks
Four biological networks were constructed from different sources of interactions between human genes or proteins. A network of physical binary protein–protein interactions was created from the PSICQUIC portal (*Del-Toro et al., 2013*) and the CCSB Interactome database (*Rolland et al., 2014*), and contains 60,669 direct interactions between 12,110 proteins. A co-expression network was constructed from RNA-Seq data downloaded from the Human Protein Atlas (http://www.proteinatlas.org). Spearman correlations of expression were computed between all genes based on FPKM values in 27 tissues and 44 cell lines, and correlation $\geq 0.7$ were selected as interactions (1,107,547 interactions between 9,212 genes). A pathway network was constructed from Biocarta (http://www.biocarta.com), Spike (*Paz et al., 2011*), Kegg (*Kanehisa et al., 2008*), PID (*Schaefer et al., 2009*) and Reactome (*Croft et al., 2014*), using the R package *graphite* (*Sales, Calura & Romualdi, 2014*). It contains 166,761 interactions between 8,839 genes. A network

of complexes was constructed from the CORUM database (*Ruepp et al., 2009*) using the matrix model, and contains 36,762 interactions between 2,528 genes. For detailed protocol and options, the markdown is available at GitHub (https://github.com/gilles-didier/MolTi). Data were fetched and networks created in May, 2015. Networks in the Fig. 4 are represented with Cytoscape (*Saito et al., 2012*).

The 4 biological networks correspond to the different layers of the multiplex biological network, sharing the same set of vertices (i.e., the complete set of human genes or proteins, considered here equally). Each layer is incomplete and provides information for a subset of genes/proteins only, the information for the remaining nodes being unknown, and therefore considered as missing data.

## Assessment of the detected communities
### Adjusted Rand index

The comparison between community partitions is performed by computing the adjusted Rand index (*Santos & Embrechts, 2009*), a widely used measure of the similarity between two partitions. The closer two community partitions, the greater their adjusted Rand index.

### Annotation enrichment

The actual biological communities are unknown, but the accuracy of the detected communities can be assessed by verifying their consistency with known biological processes. We performed exact Fisher tests (*Fisher, 1954*) to search for enrichment in Gene Ontology Biological Process (GOBP) terms (*Ashburner et al., 2000*) associated with the genes/proteins of each community. The Bonferroni correction for multiple testing was applied on the Fisher $p$-values by taking into account both the number of tested ontology terms and communities. Similarly, the communities were screened for enrichment in disease genes extracted from the CTD database (*Davis et al., 2014*) and filtered to select the curated gene-disease associations (marker/mechanisms/therapeutic).

### *MolTi* software

We developed *MolTi*, a standalone user-friendly graphical software. *MolTi* detects communities from multiplex networks provided by the users, by optimizing the multiplex-modularity with the adapted Louvain algorithm. *MolTi* further optionally performs Fisher tests of annotation enrichment for the detected communities, and allows an easy exploration of the results. *MolTi* and corresponding source code are freely available at Github (https://github.com/gilles-didier/MolTi).

## RESULTS
### The multiplex-modularity recovers accurate communities from simulated multiplex networks

We used stochastic block models (SBMs) to randomly simulate multiplex networks with a reference community structure, and composed of 1–9 graph layers (Methods, Section 'Random graph models'). To reflect some of the characteristics of biological networks in the simulated networks, we generated multiplex networks composed of sparse, dense and

mixed graphs. In addition, the incompleteness of biological networks was simulated by node withdrawal.

We chose the Louvain algorithm that optimizes the network modularity (*Blondel et al., 2008*) to identify communities. The multiplex-modularity is defined as the sum of the individual network layer modularities with regard to the same community partition (Methods, Section 'Multiplex-modularity'). We adapted the Louvain algorithm to optimize this multiplex-modularity (Methods, Section 'Clustering algorithms for community detection'). The Louvain algorithm was directly applied to the multiplex networks aggregated through their union, intersection or sum (Methods, Section 'Clustering algorithms for community detection'). Finally, we also computed a consensus clustering of the communities detected on individual networks (Methods, Section 'Consensus clustering'). The adjusted Rand index, which measures the similarity between two partitions (Methods, Section 'Adjusted Rand index'), was used to compare the communities identified with the different approaches to the reference community structure used to generate the networks.

### The intersection and union aggregations, and consensus clustering, fail to recover the community structure

We first observed that the identification of communities from the intersection of the graphs composing the multiplex network eventually leads to detect less accurate community structures (Fig. 1). An exception occurs for the simulation of dense graphs, for which we observed an improvement with the intersection of multiplex networks composed of 2 and 3 graph layers (Fig. 1C-1). In all others cases, the accuracy of the communities detected by the intersection aggregation decreases with the number of graphs composing the multiplex networks. Indeed, as the probability of observing an edge between two given vertices in all the graphs decreases with the number of graphs, the intersection tends to be empty.

A similar behavior is expected for the identification of communities from the union of the graphs composing the multiplex network, as the union aggregated network tends to be complete. This is observed for the simulation of dense and mixed-densities multiplex networks (Figs. 1C-1,2 and 1B-1,2). However, the adjusted Rand index increases with the number of graph layers on sparse simulations (Fig. 1A-1, 2). This unexpected behavior comes from the fact that these simulated networks are so sparse that their unions do not saturate, given the number of graphs considered here. In summary, both the union and intersection aggregations ultimately lead to lose the community structure.

The performance of the consensus clustering increases slowly when considering more than 2 graph layers, showing that a general community structure may eventually emerge from that detected on the individual graphs (Figs. 1A-1, 1B-1 and 1C-1). However, the accuracy of the consensus clustering never outperformed nor got close to the Multiplex-modularity accuracy in our simulations.

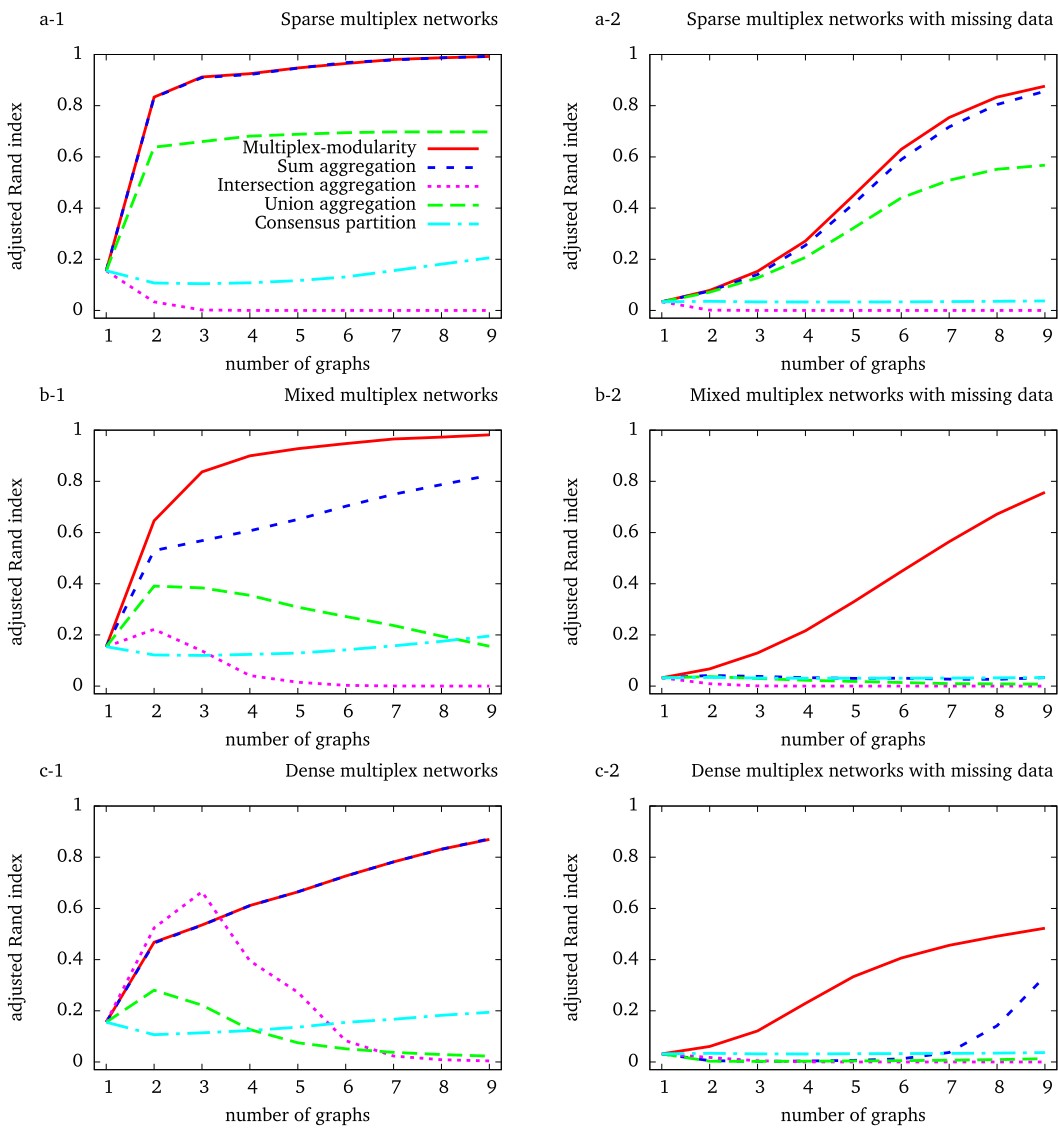

**Figure 1** **Adjusted Rand indexes between the reference community structure used to generate the random multiplex networks and the communities detected by the different approaches.** Multiplex networks contain from 1 to 9 graph layers. The indexes are averaged over 1,000 random multiplex networks of 1,000 vertices and 20 balanced communities. Sparse (resp. dense) multiplex networks are simulated with 0.1/0.01 (resp. 0.5/0.2) internal/external edge probability matrix. Mixed multiplex networks are simulated by uniformly sampling among these two matrices. Each vertex is withdrawn from each graph with a probability 0.5 to generate missing data.

## The multiplex-modularity better recovers communities in heterogeneous density and missing data contexts

In our simulations, the accuracy of the communities detected by the sum aggregation and the multiplex-modularity approach is consistently better than the accuracy of the union and intersection aggregations, or the consensus clustering. Importantly, the accuracy of the communities detected by the sum aggregation and the multiplex-modularity approach increases with the number of graph layers considered. Although the community structure

seems undetectable from a single graph, it is almost perfectly inferred by the multiplex-modularity approach when a sufficient number of graphs is reached. For instance, communities are recovered with Rand Indexes over 0.9 with 3 graphs for sparse multiplex networks (Fig. 1A-1), 4 graphs for mixed multiplex networks (Fig. 1B-1) or 9 graphs for dense graphs and sparse multiplex networks with missing data (Figs. 1C-1 and 1A-2).

The sum aggregation and multiplex-modularity approaches show very similar performances on sparse and dense homogeneous multiplex networks (Figs. 1A and 1C). However, these two approaches do differ for mixed-densities multiplex networks and when missing data are simulated to account for network incompleteness (Fig. 1, Methods, Section 'Network incompleteness and missing data'). In these cases, the accuracy of the multiplex-modularity approach is better than the sum aggregation. As expected, in a missing data context, the community structures are more difficult to detect. All the approaches based on the aggregation of layers composing a multiplex network are sensitive to missing data. Nonetheless, the multiplex-modularity approach still identifies adequate communities when the number of graphs is sufficiently large, and is particularly more accurate than the sum aggregation approach. Additionally, we compared our multiplex-modularity approach to GenLouvain, a general approach to detect communities from multiplex networks incorporating inter-slice connections (Mucha et al., 2010). In most simulated cases, GenLouvain has a behavior similar to the the sum aggregation approach (Fig. S1).

## Multiplex-modularity identifies relevant modules from a biological multiplex network
### The identification of communities from 4 biological networks and the issue of the resolution parameter

We constructed 4 biological networks from physical and functional interaction data: a protein–protein interaction network, a network of pathways, a network of protein complex, and a mRNA co-expression network (Methods, Section 'Biological multiplex networks').

These networks are sparse, heterogeneous in densities and incomplete. Their densities range from 0.001 to 0.023, with an average of 0.01. They are also prone to missing data as their sizes range from 2,528 to 12,110 nodes (Table S1). Missing data is indeed an important feature in biological networks since insufficient depth and coverage of the interaction space leads to incomplete interaction datasets (Venkatesan et al., 2009; Braun et al., 2009).

Altogether, the 4 biological networks contain 17,003 nodes and 1,371,739 edges, among which 1,338,086 appear in 1 network only (Table S1). The Louvain algorithm adapted to the multiplex-modularity was applied to identify communities from this biological multiplex network. The classical Louvain algorithm was applied to the 4 networks individually, and to their union, and sum aggregations. Considering their weak performance in the simulations, the intersection-aggregation and consensus clustering approaches were not taken into account in the following analyses.

As some of the biological networks considered here are very large, we used the resolution parameter $\gamma$ to reduce the sizes of the detected communities (Methods, Section 'Resolution parameter'). We tested the clustering of the 4 individual networks, sum and union aggregations, and multiplex-modularity approach with $\gamma$ parameter values ranging from 1 to 15. We next elected the parameter value $\gamma = 5$ that gives module average sizes lower than 50 proteins for the sum, union and multiplex partitions (Fig. S3). The number of modules obtained from each individual network ranges from 85 communities for the Pathway partition to 1,320 communities for the Co-expression network partition (Fig. S2). Using this value of resolution parameter, more than 800 communities are obtained from the sum and union aggregation networks, and 350 from the multiplex-modularity approach. The communities obtained with the multiplex-modularity approach display a median size of 17 proteins per community as opposed to a median sizes of 2 proteins per community for the sum or union aggregations (Fig. S4). The multiplex-modularity communities are hence more balanced and easier to interpret biologically.

### The multiplex-modularity clustering identifies communities closer to the individual partitions

We computed the adjusted Rand Indexes between the community partitions obtained from the 4 individual networks and from their sum and union aggregations, and the multiplex-modularity approach (Methods, Section 'Adjusted Rand Index'). We observed that, on average, the multiplex-modularity partition is closer to the individual network partitions than the aggregations (Fig. 2).

### The multiplex-modularity identifies communities enriched in gene ontology terms

To assess the consistency of the communities, we tested their enrichment in Gene Ontology Biological Process (GOBP) terms (*Ashburner et al., 2000*, Methods, Section 'Annotation enrichment'). The communities detected by the multiplex-modularity approach are more often associated to at least one significant ($q$-value $< 0.05$) GOBP term than the sum or union aggregation communities (Fig. 3). For $\gamma = 5$, for instance, 38% of the multiplex-modularity communities are annotated, compared to 14% and 10% of the sum and union communities. Overall, these results suggest that the multiplex-modularity approach is extracting relevant biological communities.

### Clustering of proteins involved in the Coffin-Siris syndrome and other syndromes with overlapping clinical features

As an illustration, we screened the modules obtained with the multiplex-modularity approach ($\gamma = 5$) for enrichment in disease-related proteins (Methods, Section 'Annotation enrichment'). Twenty-nine communities are significantly associated to at least one disease. Among those, one module contains the 6 gene products known to be associated to the Coffins-Siris syndrome ($q$-value $= 5 \times 10^{-10}$, Fig. 4, Methods, Section 'Annotation enrichment'). This syndrome is a rare congenital disorder characterized by specific cranofacial and digital features, microcephaly and intellectual disability, among other phenotypes (*Kosho & Okamoto, 2014*; *Vergano & Deardorff, 2014*).

a

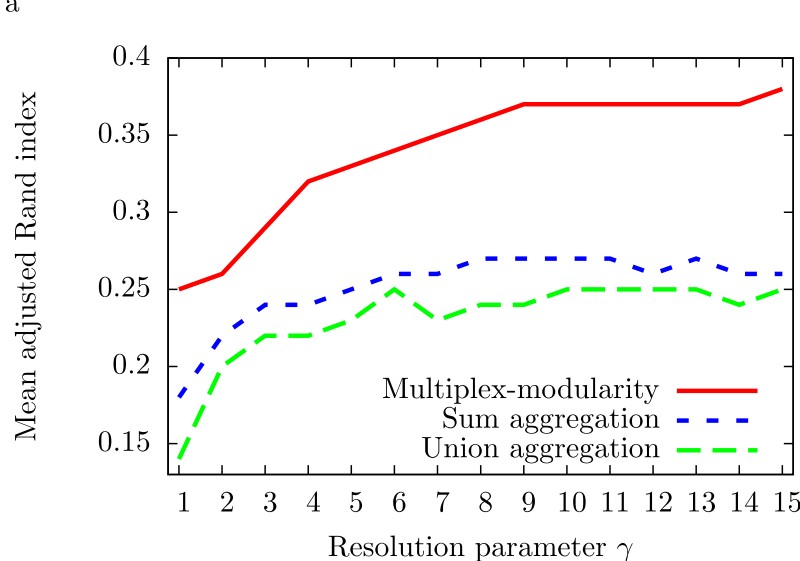

b

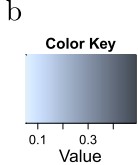

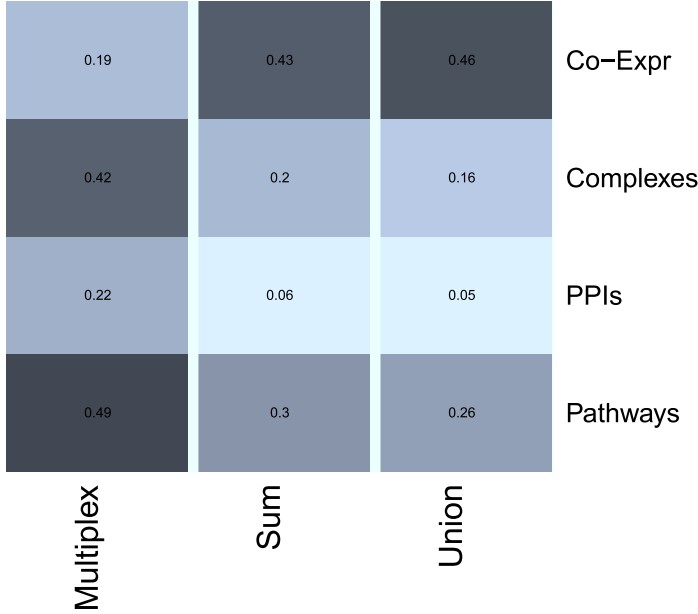

**Figure 2** **Adjusted Rand Indexes between the partitions of the 4 individual networks and their sum-aggregation, union-aggregation, and the multiplex-modularity approach.** (A) Mean adjusted rand indexes for $\gamma = 1$ to $\gamma = 15$. (B) detailed adjusted rand indexes for $\gamma = 5$.

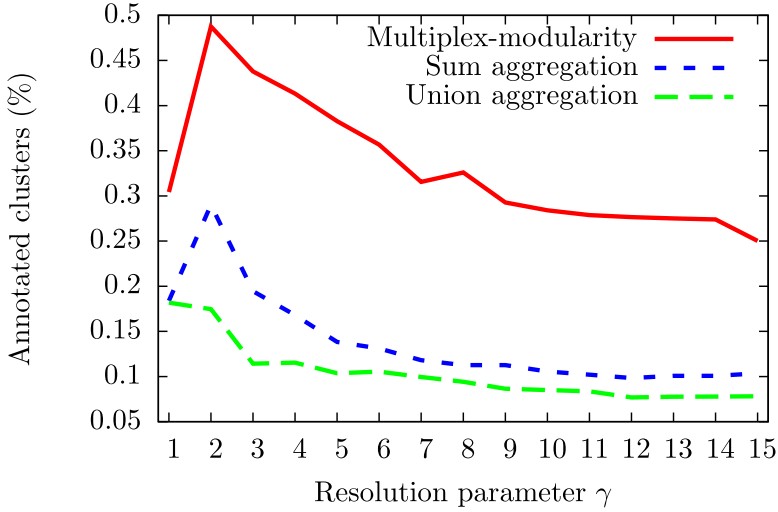

**Figure 3** Percentage of communities associated to at least one significant GOBP term, for $\gamma = 1$ to $\gamma = 15$.

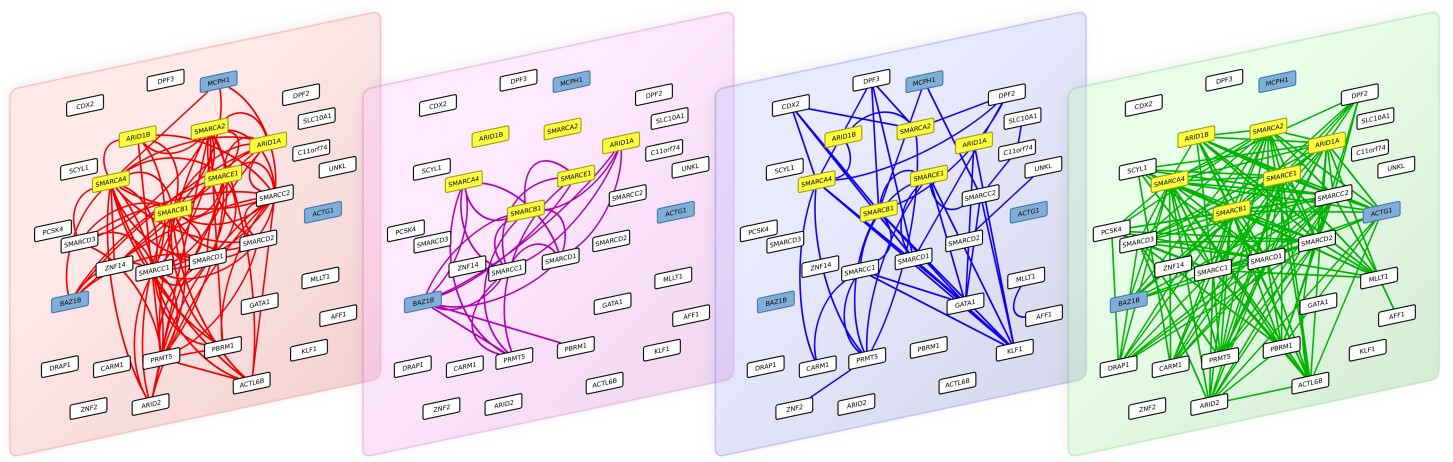

**Figure 4** The 4 interaction layers of a module obtained after partitioning the multiplex biological networks with the multiplex-modularity approach. From left to right: pathways, Co-expression, PPIs and Complexes networks. Proteins involved in the Coffin-Siris syndrome are highlighted in yellow, and protein related to other syndrome with shared clinical features are highlighted in blue.

The module is enriched in proteins involved in transcription regulation, and participating in different cellular processes such as chromatin remodeling, transcription from RNA-polII promoter and DNA repair. The 6 Coffin-Siris syndrome proteins are components of the SWI/SNF chromatin remodeling complex (*Tsurusaki et al., 2012*). Interestingly, the module also contains other gene products implicated in other syndromes, which clinical features overlap with the Coffin-Siris syndrome. MCPH1 is involved in primary autosomal recessive microcephaly (*Venkatesh & Suresh, 2014*), BAZ1B is involved in the William-Beuren syndrome that also presents neuropsychological deficits (*Fusco et al., 2014*; *Xiao et al., 2009*) and ACTG1 mutations are involved in the Baraitser-Winter syndrome, another developmental disorder associated to intellectual disability

(*Rivière et al., 2012*). It is to note that all these syndrome-related proteins are clustered only when the multiplex-approach is used. Indeed, when the different network layers are aggregated, only 4 out of the 6 Coffin-Siris syndrome proteins belong to the same module.

## DISCUSSION

Interactions between genes and proteins can be of various nature and are identified by diverse experimental approaches. We focused here on different facets of protein cellular functions by taking into account (i) physical interactions between proteins, extracted from experiments such as yeast 2-hybrid screens, (ii) interactions derived from the belonging of proteins to complexes, (iii) functional interactions extracted from pathway databases and (iv) co-expression correlations derived from mRNA expression in different cellular contexts. However, other types of functional relationships between genes or proteins, such as their implication in the same disease (*Goh et al., 2007*) or their targeting by the same drug (*Yildirim et al., 2007*) could be considered. Consequently, many networks can depict gene/protein functional relationships. Each of them represents a different layer of a multiplex biological network, and each layer has its own meaning, size, relevance and bias. For instance, the network of pathways is derived from information provided by several pathway databases containing heavily curated expert knowledge rather than large-scale datasets prone to noise. Conversely, the network of correlations computed from RNA-seq expression data contains a lot of links, some of which are undoubtedly noise. Furthermore, the different experimental approaches sourcing the interaction data capture different parts of the interaction space, therefore leading to incomplete networks. It is estimated for instance, that only 20% of the human protein–protein interaction space has been currently deciphered (*Menche et al., 2015*). Similarly, curated interactions in pathway databases are limited to the proteins for which such information exists, letting aside a potential wealth of data yet to be deciphered. In this context, taking into account diverse network sources when deriving functional modules is expected to cover a more comprehensive picture of protein cellular functioning. The underlying reason is to combine informative-but-sparse and plenty-but-noisy information to strengthen each other, and improve the clustering performance (*Papalexakis, Akoglu & Ience, 2013*).

We explored here different approaches based on modularity to identify communities from a set of networks. These approaches are all based on the assumption that the layers of a multiplex network have the same community structure. For biological networks, this means that the different categories of gene/protein functional relationships are considered as instances or realizations of the real underlying functional modules. In our work, this assumption is also fulfilled by the simulated random multiplex networks, as they are generated from the same reference community structure. The aggregation of the network layers into a monoplex network, or the consensus of communities are classically applied to biological networks. The use of a natural extension of the modularity metric to multiplex networks is newer (*Bennett et al., 2015*, this work).

While performing these analyses, *Bennett et al. (2015)* published an extension of the modularity to multiplex networks equivalent to our multiplex-modularity. The originality

of our work lies in the extensive comparisons we performed with the aggregation and consensus approaches. In particular, we used random simulated networks to precisely study the effects of different network topological features. This is particularly important for the identification of communities from multiplex biological networks, as their density and incompleteness can vary. Overall, when simulating networks with heterogeneous densities and missing data, the multiplex-modularity approach outperforms all its aggregated counterparts. This better performance is also observed in a real-case multiplex biological network, for which the multiplex-modularity approach identifies more balanced and annotated communities.

Other clustering approaches on multiplex networks were not considered because they cannot readily be applied to detect communities from large-scale multiplex biological networks with missing data (*Shiga & Mamitsuka, 2012*; *Papalexakis, Akoglu & Ience, 2013*). The GenLouvain approach (*Mucha et al., 2010*), a generalization of the modularity to any kind of multi-slice networks, requires parameters difficult to set for biological networks, such as inter-layer coupling (*Mucha et al., 2010*). Hence, we applied GenLouvain only to the simulated multiplex networks (Fig. S1).

The large-scale interaction networks available nowadays constitute both a fantastic source of information to study proteins functioning in their cellular context, and a huge challenge calling for the development of methods to mine knowledge from such complex data. We established here that the multiplex framework is suited to combine interaction networks of different nature and allows recovering relevant functional modules.

## ACKNOWLEDGEMENTS

We thank many members of our social and professional multiplex network for their kind reviews and discussions.

### Funding

This work was supported by the Centre National de la Recherche Scientifique (PEPS BMI IMFMG) and the A*MIDEX project (no ANR-11-IDEX-0001-02) funded by the ''Investissements d'Avenir'' French Government program, managed by the French National Research Agency (ANR). The funders had no role in study design, data collection and analysis, decision to publish, or preparation of the manuscript.

### Grant Disclosures

The following grant information was disclosed by the authors:
Centre National de la Recherche Scientifique.
Investissements d'Avenir: ANR-11-IDEX-0001-02.

### Competing Interests

Anaïs Baudot is an Academic Editor for PeerJ. The authors declare there are no competing interests.

## Author Contributions

- Gilles Didier and Anaïs Baudot conceived and designed the experiments, performed the experiments, analyzed the data, contributed reagents/materials/analysis tools, wrote the paper, prepared figures and/or tables, reviewed drafts of the paper.
- Christine Brun conceived and designed the experiments, analyzed the data, wrote the paper, reviewed drafts of the paper.

## Data Availability

https://github.com/gilles-didier/MolTi.

## Supplemental Information

Supplemental information for this article can be found online at http://dx.doi.org/10.7717/peerj.1525#supplemental-information.

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
