# Peer review of "Identifying communities from multiplex biological networks"

_PeerJ, doi:10.7717/peerj.1525_

## Round 0.1 · original submission · Major Revisions

Please take time to address the major concerns raised by Reviewers 1 and 3.

Reviewer 1 ·

Basic reporting

I agree that the comparison among the basic methods for detecting communities from multiple networks is very important and useful and might not be done so far. So the idea of this work might be on the right track, while I'd like to raise the following problems of this work.

1. I am unable to satisfy with the current introduction on the past methods in the literature. I think the corresponding part in Introduction should be written more properly.

More concretely, the authors say that the methods can be classified into three sets of approaches, while I am unable to see what type of assumptions behind each of the three sets. That is, for example, I think the simplest assumption would be that communities should be shared by all given multiple networks. (This might be an idea of consensus clustering.) A bit more complex assumption is that each community can be shared by only one or part of all given networks. This case a possible primitive method would be to capture communities from each of given networks and then combine them over all networks, while this case the question is how the conflict of different communities should be fixed.

I think first of all this type of assumptions of existing methods should be clarified. Then the methods should be categorized due to their assumptions on communities.

2. The authors define multiplex-modularity as the sum of modularities of networks, while no validation on this definition nor any motivation is given. Also no comparison with other methods. (The authors mention "natural extension" but I think this is a "simple" or "ad hoc" extension to multiple networks. They should explain the reason why this is valid. In other words, what is the assumption behind this definition? And why the assumption is valid? This should be explained in the method section. In Discussion the authors mention that "This multiplex modularity is based on the assumption that all the layers of a multiplex network have the same community structure." Is this not clear enough but even so correct? This sounds that multiplex modularity is similar to the simplest assumption I mentioned above, and then this assumption is the same as consensus clustering basically. So why is the performance different from consensus clustering? Also I am unable to agree that this assumption is good for multiple biological networks, because biological networks such as PPI and metabolic networks are derived from different biological phenomena, meaning that they are not necessarily sharing the same community structures always.

As I mentioned in the first comment, I think any proposed method so far must have some assumption, while in this work, the assumption behind the proposed multiplex modularity is not clearly validated enough, by which readers are unable to see what kind of communities can be detected by this definition.

Experimental design

The authors defined the mulitpex-modularity in an ad hoc manner, and compared this definition with very naive approaches like sum, intersection and union. Intuitively it is very clear that intersection and union are really weak for many situations. Also consensus clustering. In other words, we can see the reason why these methods are weak under what conditions. On the other hand, the readers are unable to see why multiplex modularity works under what condition. The main reason for this is I think the multiplex modularity is defined ad hoc, and no validity is shown. Again I think the authors should clarify why multplex modularity is valid and work under what conditions. Also they should compare the multiplex modularity with already existing methods for multiple networks, particularly those proposed for a similar purpose. Otherwise it is very hard to evaluate the proposed method.

Validity of the findings

In light of the above comments, I think the experimental results are unfortunately not worth enough to make the findings scientifically worth. Again this was caused by the following two reasons: 1) multiplex modularity is defined ad hoc and the authors should show what can be realized by this criterion clearly and 2) the empirical comparison is not rigorous enough. The authors should compare the proposed method with more recent methods for the same or similar purpose.

Additional comments

To summarize, the authors' idea/target of the defined multiplex modularity was not shown properly. This is also due to the authors' understanding of the past methods, which are also rather superficial and not like capturing the idea/assumption behind the reason why those methods were proposed. I recommend that this point should be fixed in this work.

1. The clustering/detecting communities algorithms which were used in experiments are not clear enough. Their procedures should be shown more in detail.

I guess the difference of consensus clustering and the defined multiplex modularity may be caused by the difference of algorithms, because the assumption behind these approaches is basically the same.

2. Again the assumption of the multiplex modulairty mentioned in Discussion: "This multiplex modularity is based on the assumption that all the layers of a multiplex network have the same community structure." is vague.

I'm thinking that eventually there might be TRUE community structures, while they are not shared by all networks and part of them appear in each network. In fact, from Figure 4, you can see Co-expression and PPI show totally different networks, not sharing with each other (even assuming that they are so noisy). So I think the method for clustering (detecting communities) for multiple networks should be the method which can capture different communities from each network and keep the communities which have rather low conflicts among given networks. Anyway once again, I'd like to ask the authors what they want to have from multiple networks and if they can be realized by multiplex modularity.

Reviewer 2 ·

Basic reporting

No comments.

Experimental design

No comments.

Validity of the findings

No comments.

Additional comments

In this manuscript, the authors introduce the notion of multiplex-modularity and present a comparison among some methos for module extraction from biological networks, based on that measure. I found the paper easy to read and well organized.

Reviewer 3 ·

Basic reporting

The authors address the problem of community detection in multiplex networks. This is a timely subject that has received considerable interest recently. Overall, although the subject is interesting, the paper is poorly written and hard to follow. I have the following criticisms (in the sections below)

Experimental design

Major comments
1. It is claimed that “the Louvain algorithm was adapted to find the community partition optimizing the multiplex-modularity", however it is not clear how the procedure for adapting the algorithm was implemented. The Louvain method has already been extended to handle multiplex networks in the genLouvain algorithm, with modularity optimisation that is applicable to any number of networks slices of any interaction type (Mucha et al. - Science 328:876–878, 2010). No effort is made in the manuscript to explain the differences between this method and the genLouvain algorithm and no comparison with this method is provided. A consensus algorithm "inspired from Lancichinetti and Fortunato" is mentioned, but it is poorly described (only 5 lines of text).
2. With regards to attempts of describing the functional content of communities, the authors state that they find the modules with "at least one significant GO BP" (Fig. 3). However, this process does not provide enough detail about the overall performance of the method, which would be very useful when comparing different methods. What about a method that finds less annotated modules that contain more GO BP terms? Why should it be worse than one that finds more modules with “at least one” enriched GO BP term? In addition, it is not clear if the GO enrichment method used takes into account the inheritance problem (Bauer et al. - Bioinformatics 24:1650–1651, 2008), i.e. that a gene which is annotated to a GO term is also annotated to all of its parent (less specific) terms. Furthermore, only a comparison between aggregated networks is provided (which are expected to have worse performance) and no comparison with other consensus or multiplex methods is being made (Fig. 3).
3. Biological significance is discussed with respect to the Coffin-Siris syndrome (Fig. 4). Although an interesting example, it constitutes only one module within the entire network and a more comprehensive assessment of performance across all modules in the entire network would be required. As noted, “all these syndrome-related proteins are clustered only when the multiplex-approach is used. Indeed, when the different network layers are aggregated, only 4 out of the 6 Coffin-Siris syndrome proteins belong to the same module”. The argument that the multiplex approach finds 2 more proteins in a single module is valid but is a weak indication to support the general conclusion that multiplex modularity is making a significant improvement in the discovery of meaningful biological modules.

4. A Stochastic Block Model (SBM) with fixed probabilities for internal and external edges has been used, where all communities will have the same size. This is not realistic realistic for representing the behavior of real biological networks. Perhaps other alternatives (LFR benchmark) which incorporate communities of different sizes preferable.

5. The selection of gamma (gamma = 5) parameter for the real life networks needs to be justified.

6. Language and clarity of expression should be improved throughout.

Other comments:
As mentioned in the abstract and the manuscript "a user-friendly graphical software to detect communities from multiplex networks is available", however the software has a command line and not a graphical interface, therefore can be deemed as not as user friendly as claimed.

There are no labels for the axes in Fig.1. While this is described in the caption, including them in the figure would make it much easier to read.

Equations are not numbered.

Validity of the findings

Please see above

---

## Round 0.2 · accepted · Accept

I apologize for the length of time for a decision. This was in part due to delays in receiving reviews as well as time to consider reviewer responses and downstream discussions regarding the manuscript.

As you are aware, there have been a range of reviewer responses, ranging from accept to reject, with the latest revision having a reviewer change their response from “major revisions” to “reject”.

Following considerable discussion it was thought that there was a good faith effort to address the reviewers’ concerns, including a comparison to the GenLouvian method using simulated data included in the supplemental materials. Given this and other components of the response, the decision is to accept your manuscript for publication.

Reviewer 3 ·

Basic reporting

No comments

Experimental design

In the original review of the manuscript, I mentioned that comparison with the GenLouvain method is required. The authors have replied that 'The GenLouvain method is available in MathLab but requires computing skills' which is not a valid reply on why this benchmark was not attempted, especially as it is an important part of validating any conclusions and method reporting.

Validity of the findings

Findings are week to merit publication. Comparison with methods, such as GenLouvain, is key to validate findings.